# TabText: A Flexible and Contextual Approach to Tabular Data Representation

## Abstract

Tabular data is an essential data format for applying machine learning tasks across various industries. However, traditional data processing methods do not fully utilize all the information available in the tables, ignoring important contextual information such as column header descriptions. In addition, pre-processing data into a tabular format can remain a labor-intensive bottleneck in model development. This work introduces TabText, a processing and feature extraction framework that extracts contextual information from tabular data structures. TabText addresses processing difficulties by converting the content into language and utilizing pre-trained large language models (LLMs). We evaluate our framework on ten healthcare prediction tasks—including patient discharge, ICU admission, and mortality—and validate its generalizability on an additional task from a different domain. We show that 1) applying our TabText framework enables the generation of high-performing and simple machine learning baseline models with minimal data pre-processing, and 2) augmenting pre-processed tabular data with TabText representations improves the average and worst-case AUC performance of standard machine learning models by as much as 5% additive points. All the code to reproduce the results can be found at `https://anonymous.4open.science/r/TabText-18F0`.

## 1 Introduction

Tabular data remains the most widely used and readily available data format across various fields ranging from education, healthcare, and technology, where it serves a vital role in capturing all domains of information. Pre-processing tabular data accurately and efficiently is essential for creating reliable downstream models in machine learning applications. Yet, two significant limitations exist for directly incorporating tabular data into generalizable modeling pipelines: they require labor-intensive, often manual, data processing to standardize information across heterogeneous tabular structures and data sources, and they ignore contextual information such as column headers and meta content descriptions.

In contrast to tabular approaches, language is a very flexible data modality that can easily represent information about different data points without imposing any structural similarity between them. Furthermore, recent developments on off-the-shelf large language models (LLMs) based on the Transformer architecture (Vaswani et al., 2017) offer state-of-the-art performances on a wide range of language tasks, including translation, sentence completion, and question answering. These pre-trained models are often developed with very large and diverse data sets, allowing them to exploit prior knowledge and make accurate predictions with very few new training samples. Some LLMs are trained to target specific domain knowledge and technical challenges, making them particularly useful in the corresponding applications. For example, LLMs fine-tuned on clinical notes and biomedical corpora such as ClinicalBERT (Alsentzer et al., 2019), BioBERT(Lee et al., 2019), and BioGPT(Luo et al., 2022) offer substantial advantages for medical learning tasks, and LLMs that specifically target long token sequences unveil opportunities for dealing with data that contains long texts (Beltagy et al., 2020; Li et al., 2022).

These successful language models offer a natural solution to represent and process contextual information from tabular structures. Standard machine learning models only utilize the explicit table contents, disre-

garding all accompanying context like column headers and table descriptions. Incorporating these metadata into the model via language could give meaning to the data values within the broader context. For example, a numerical value might be very relevant for disease prediction if it represents a person's age but not so much if it corresponds to the ward census. Moreover, LLMs could save significant manual labor for selecting, encoding, and imputing data (Sweeney, 2017; Geneviève et al., 2019; Nan et al., 2022). Missing data, in particular, is a challenging and frequent problem that requires attentive processing and expert knowledge. Current predictive models either exclude such attributes, potentially ignoring rare-occurring but valuable data or impute missing values with very few recorded instances. Additional processing challenges arise when units of measurement or data types are inconsistent across tabular data systems. By leveraging language, these difficulties could be addressed, for instance, by simply writing that particular values are missing and converting inconsistent values into text.

Recent advances in tabular deep learning span architectural, training, and representation-level innovations. Foundation models like TabPFN and TabForest (Hollmann et al., 2025; Breejen et al., 2024) apply in-context learning to tabular data; enabling flexible, task-adaptive predictions. Other works explore hybrid approaches, such as nearest-neighbor-enhanced deep models (Ye et al., 2024; Gorishniy et al., 2023), or leverage modern deep learning architectures designed specifically for tabular data (Gorishniy et al., 2021; Thielmann et al., 2024; Gorishniy et al., 2024; Huang et al., 2020). In this work we focus on a preprocessing-based approach that converts tabular rows into textual formats, enabling improved performance for classical machine learning models like gradient-boosted trees via richer, semantically informed feature embeddings.

Previous works using LLMs have shown the potential of using natural language processing (NLP) models to systematically and efficiently process tabular data in the form of language (Herzig et al., 2020; Wang et al., 2023; Padhi et al., 2021). However, they have mainly relied on training fixed BERT-based models that are not flexible to changes in tabular structures. These works have mostly assumed that encoding data using LLMs leads to better performance than traditional data processing methods, but concrete evidence has not been provided. Other works augment tabular data with external unstructured data (Harari & Katz, 2022) but do not leverage contextual data from the original tabular source. In addition, language models are considered sensitive to their input representations (Miyajiwala et al., 2022), and most previous works do not thoroughly investigate how the choice of language affects their results. Hegselmann et al. (2023) investigate different language variants, but in the context of zero-shot and few-shot classification as opposed to feature representation. Wang et al. (2023) incorporate contextual information derived from input tables using LLM-generated prompts; however, they do not analyze how variations in the generated text affect downstream model performance. Thus, guidance on the best way to construct the language data remains in need.

This paper builds and evaluates a new feature extraction methodology and addresses the questions above. We process tabular data by creating a text representation for each data sample. We then use this text as input for a pre-trained LLM to generate TabText embeddings of a fixed dimension. Finally, we augment the tabular features with these TabText embeddings to train any standard machine learning model for downstream prediction tasks.

The main contributions of this work are as follows:

1. We develop TabText, a systematic framework that leverages language to extract contextual information from tabular structures, resulting in more complete data representations.

2. We show the effectiveness of using TabText representations for general data pre-processing. Our experiments demonstrate that augmenting tabular data with our TabText representations can improve the AUC score by up to 5% across 19 classification tasks, with larger improvements observed for harder tasks.

3. We investigate the impact of several language syntactic parsing schemes on the performance of TabText representations and demonstrate the flexibility of our framework compared with traditional labor-consuming processing of unorganized data into a tabular feature space.

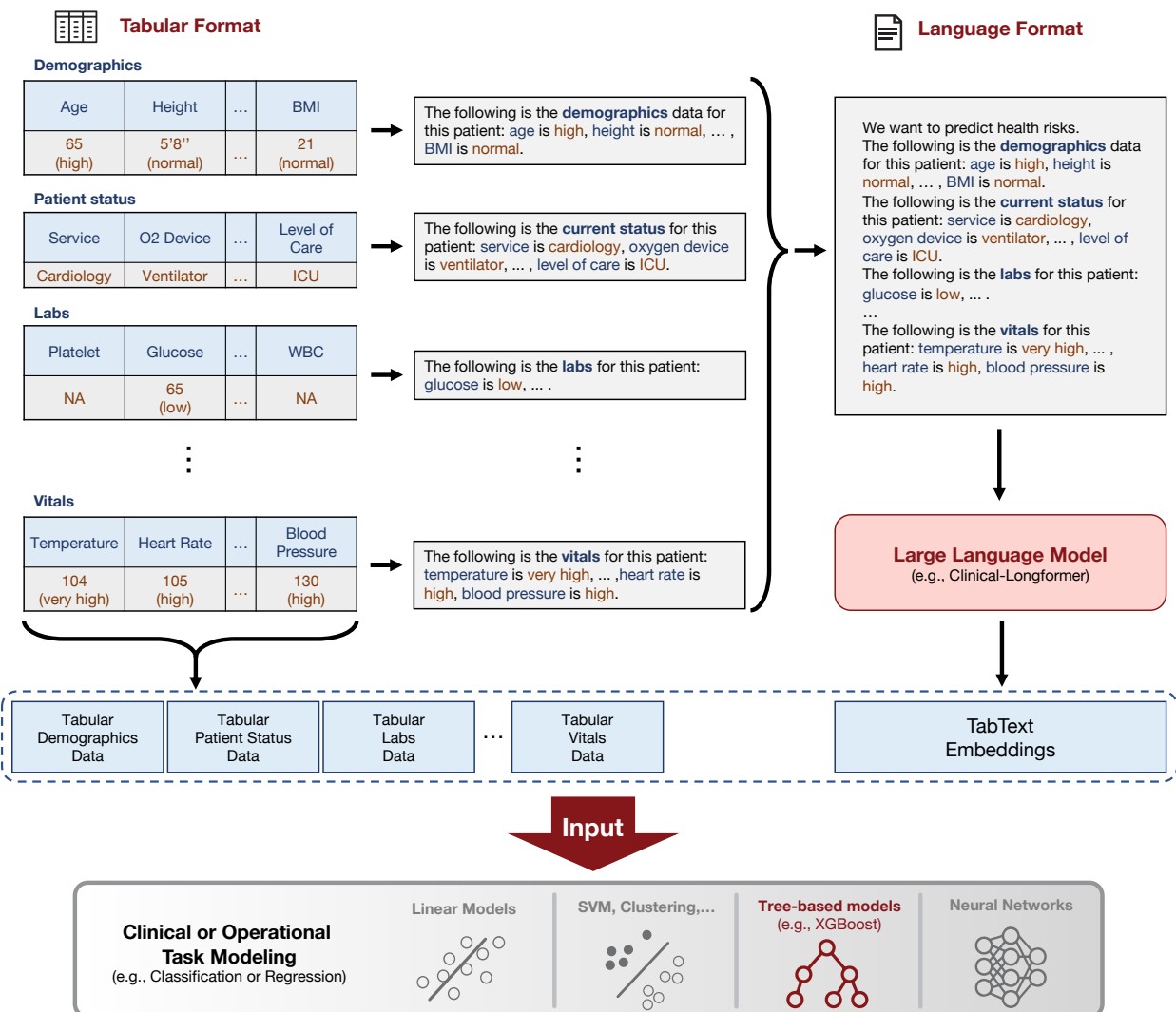

Figure 1: End-to-end TabText framework. The TabText framework integrates diverse tabular data sources cohesively into a language format that can be combined into a single patient description paragraph. This patient language representation is then fed to existing pre-trained LLMs to generate embeddings. An important advantage of this framework is the flexibility to replace this LLM as new models become available.

## 2   Methodology

Tabular data are often stored in various tabular formats. For example, in the context of healthcare, different tabular data sources may include demographics, vital signs, laboratory results, treatment status, etc. Effortless integration and utilization of these heterogeneous data for predictive modeling remains a significant challenge. The TabText framework addresses this by converting multiple structured data sources into a unified natural language representation. Specifically, it transforms each table into descriptive sentences that are then concatenated into a single summary paragraph for each data subject. This text contains the attributes of the columns with their corresponding values and potentially other available contextual information. A pre-trained LLM subsequently processes this language-based representation to generate fixed-size TabText embeddings that encode the textual data. Finally, we augment the tabular features with these TabText embeddings to train any standard machine learning model (e.g., decision trees, support vector

machines, gradient-boosted trees) for downstream prediction tasks. Figure 1 illustrates the overall TabText framework.

Given the design of our TabText framework, three main questions need to be addressed:

1. Which pre-trained LLM to use?

2. How to convert tabular data into language?

3. Should the LLM be fine-tuned with the specific language format chosen?

In this section, we answer these in the context of healthcare. However, as our experiment section demonstrates, the overall pipeline can be applied to any other domain.

In developing our methodology, our goal is to address the above questions in a manner that generalizes across diverse tasks. Specifically, we aim to generate a single TabText embedding per data subject that performs well across multiple downstream prediction tasks rather than tailoring embeddings to individual tasks. This approach not only promotes embeddings that capture all the relevant information in the input text, but also reduces computational overhead by eliminating the need to recompute embeddings for each new task.

To this end, we consider nine different binary classification tasks largely overlapping with Na et al. (2023), all related to inpatient flow predictions in a hospital. Two discharge-related outcomes are whether patients are discharged or not within the next 24 hours (resp. 48 hours). Four ICU-related outcomes include whether the patient will enter (resp. leave) the intensive care unit (ICU) for patients currently not in the ICU (resp. in the ICU) within the next 24 hours (resp. 48 hours). Two short-term expiration outcomes concern whether each patient will die in the next 24 hours (resp. 48 hours). One end-of-stay mortality outcome indicates whether patients die or not at the end of their stay.

Throughout this section we utilize a private real-world data set comprised of 63 columns corresponding to different laboratory results of inpatients in a teaching hospital. We use $60,000$ data samples for training and validation, and $10,000$ for testing, where each data point corresponds to a patient day. All results in Sections 2.1, 2.2, 2.3 utilize this data set and all nine prediction tasks mentioned above.

## 2.1 LLM Selection

We consider two different Transformer models, BioGPT and Clinical-Longformer (Li et al., 2022), both of which were pre-trained with MIMIC-III clinical notes (Johnson et al., 2016). Following the TabText framework, we convert the tabular data into simple text: for each row, the cell from column "attribute" with value X is transformed into "attribute: X" and the texts from all columns are concatenated into a single sentence with the comma character. We next create TabText embeddings and finally train gradient-boosted tree models for the nine tasks of interest. Figure 2a shows the boxplots for the out-of-sample AUC over 10 random 75%-25% train-validation splits for each task and each model. Both NLP models achieve similar performance across tasks, and we therefore choose the Clinical-Longformer model, as it allows for input text of larger size.

## 2.2 Language Construction

The versatility of language creates a challenge for consistency, as multiple textual expressions can convey the same information. Moreover, tabular data is often split across multiple tabular sources (e.g., vitals table, medications table), some of which include information only for a particular group of data subjects. This results in even more possibilities for textual representation.

The TabText framework creates a single paragraph for each data sample (e.g., for each patient day) as follows: we first create a sentence for each column in each table. Next, for each table, we concatenate contextual information and the sentences of its columns using the colon (":") and comma (",") characters, respectively. We then merge the text from all tables into a single paragraph using the period (".") character. While the exact punctuation doesn't significantly impact BERT-based transformers (Ek et al., 2020), the

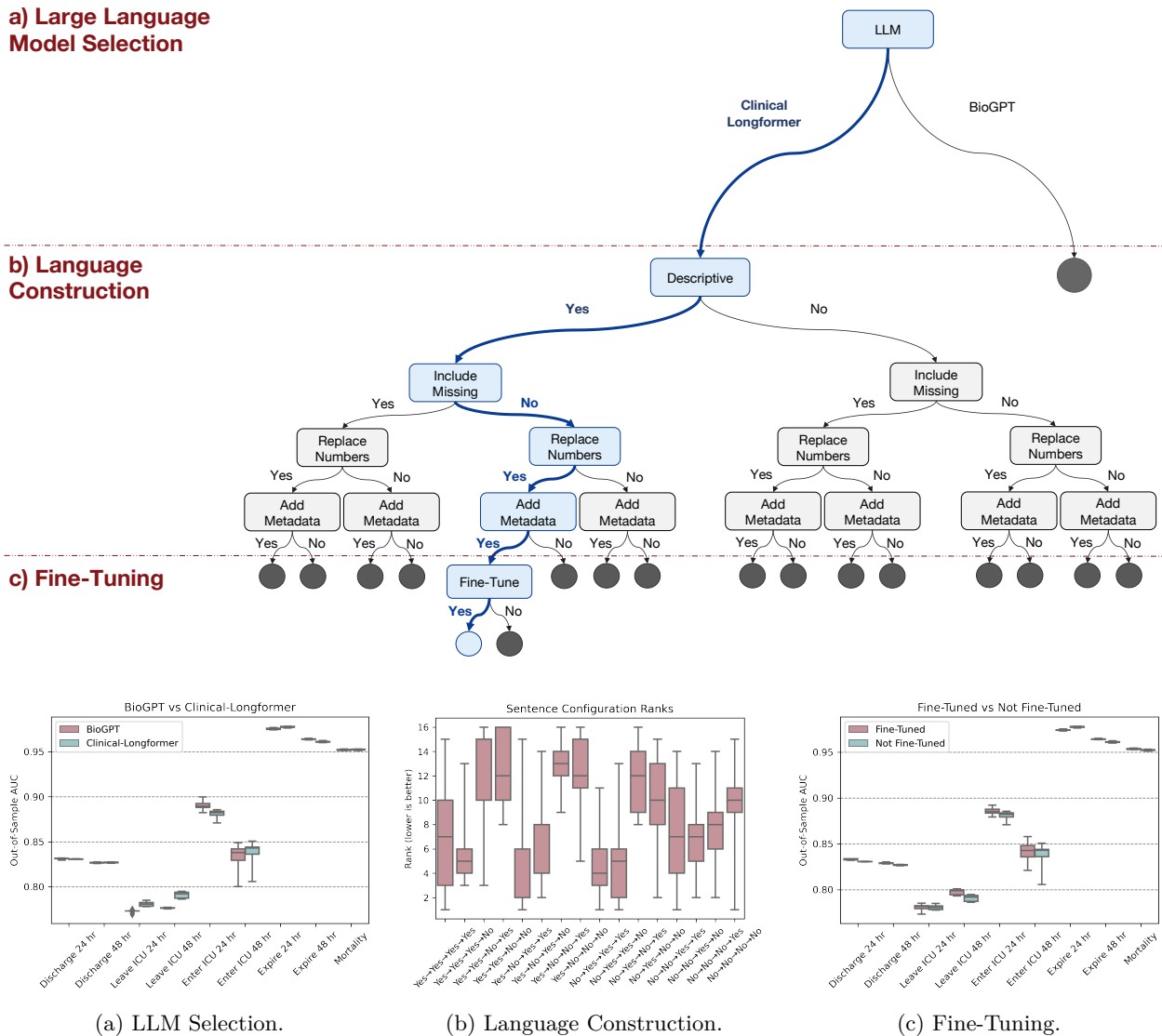

(a) LLM Selection.     (b) Language Construction.     (c) Fine-Tuning.

Figure 2: Overview of our overall methodology. We start with the selection of an LLM. Figure (a) shows that BioGPT and Clinical Longformer achieve similar results, and we therefore choose Clinical Longformer, as it allows for input texts of larger sizes. Then, we look for the best language representations of the original patient data. In Figure (b) we observe the boxplots of the ranks for different sentence configurations, which were tested across different prediction tasks (lower rank is better). We select the configuration with the lowest median ranking; specifically, we use descriptive language, omit missing values, replace numerical values with text, and include metadata. Lastly, we fine-tune the LLM using this sentence configuration, as it leads to better performance, as shown in Figure (c).

exact text chosen to build each sentence might have a larger impact on the final embedding. We therefore investigate different ways to construct sentences for each column attribute.

*Descriptiveness:* We consider whether or not to use descriptive language to construct text sentences. Specifically, consider a cell from column "attribute" that has value "X". If the column is non-binary, we consider the following options:

- Non-Descriptive Sentence: "attribute: X";

- Descriptive Sentence: "attribute is X".

For binary columns, we consider the verb associated with the specific attribute. For instance, if the column attribute is associated with the verb "to have" we consider

- Non-Descriptive Sentence: "has X: yes" or "has X: no";

- Descriptive Sentence: "has X" or "does not have X".

*Missing Values:* When the value for a column "attribute" is missing, we consider two options, to explicitly mention in the text that this information is not available ("attribute is missing"), or to simply skip this column when building the text representation.

*Numerical Data:* Discretizing continuous features has been shown to be an effective strategy for deep learning models (Gorishniy et al., 2022; 2021). Therefore, we also consider whether or not to discretize numerical values by replacing them with text. For replacement, we compute the average (AVG) and standard deviation (SD) of the corresponding column with respect to the training data. We then replace a given cell value $X$ as follows:

- "very low" if $X < \text{AVG} - 2\text{SD}$;

- "low" if $\text{AVG} - 2\text{SD} \leq X < \text{AVG} - \text{SD}$;

- "normal" if $\text{AVG} - \text{SD} \leq X < \text{AVG} + \text{SD}$;

- "high" if $\text{AVG} + \text{SD} \leq X < \text{AVG} + 2\text{SD}$;

- "very high" if $\text{AVG} + 2\text{SD} < X$.

*Including Metadata:* We investigate the added value of including metadata as part of the text representation. This corresponds to descriptions of table content (e.g., "This table contains information about the medications administered to this patient") or the prediction task of interest (e.g., "We want to predict mortality risk").

For each of the 16 possible sentence configurations, we use default values of the Clinical-Longformer model to obtain TabText embeddings that are given as input to a gradient-boosted tree model. In Figure 2b, the Language Construction results show the boxplot for the rank achieved with each configuration across tasks, where lower numbers correspond to better ranking. We choose the sentence configuration with the lowest median ranking; specifically, we use descriptive language, omit missing values from the text, replace numerical values with text, and include metadata.

### 2.3 Fine-Tuning

Although Clinical-Longformer was pre-trained with large language data sets, we can further improve its performance with a few more training iterations using our training data. Specifically, we convert our training data into language following the sentence configuration selected in Section 2.2, and we use it to fine-tune Clinical-Longformer following the original BERT training methodology, which includes self-supervised masked word prediction. We then generate embeddings that are given as input to a gradient-boosted tree model. We show in Figure 2 the boxplots for the out-of-sample AUC over 10 random 75%-25% train-validation splits for each task. We see that fine-tuning the model with our local data slightly improves performance for eight out of the nine classification tasks of interest.

## 3 Results

This section presents extensive computational experiments evaluating the performance of our TabText framework. First, we show how our pipeline can quickly generate machine-learning models with competitive performance without any data cleaning by leveraging the flexibility of language. We then demonstrate with pre-processed data that augmenting standard tabular representations with our TabText embeddings can

increase out-of-sample AUC by up to 5% additive points, with the largest improvements observed for the most challenging predictions.

*Text Encoder:* We first convert the input training data from tabular to textual format as described previously in Section 2.2. We use the sentence configuration that led to the highest average AUCs (i.e., skipping sentences for missing values, replacing numbers with text, using descriptive language, and adding metadata) to fine-tune the Clinical-Longformer model for 3 epochs. For fine-tuning we sample a random subset of 2000 points from the training set, and we keep the default values for all hyperparameters. Then, we use the fine-tuned LLM to extract language embeddings of size 768 by mean-pooling over the last hidden layer.

*Training Methodology:* For each prediction task, we compare two approaches: our TabText framework (see Figure 1) and the standard Tabular approach in which only the tabular data is given as input to the machine learning model. For all reported results, the average performance is computed over 10 random train-validation splits (identical 10 splits across all experiments) for a fair comparison. The optimal model is selected using a hyperparameter grid search (see details in the appendix) based on its performance on the validation set. We emphasize that, for the TabText framework, both the sentence construction and model fine-tuning are performed exclusively on the training set of each data split, ensuring that no information from the test set or validation set is used during training.

*Computational Cost:* Compared to the traditional tabular approach, the TabText framework incurs higher computational costs due to finetuning, embedding extraction, and training on higher-dimensional data. To keep finetuning efficient, we limited all experiments to fewer than 2,000 iterations, ensuring runtimes under 10 minutes. Embedding extraction using Longformer-based LLMs with mean pooling averaged 0.9 seconds per sample, varying with text length. For our largest dataset (800,000 samples and 160 columns), extraction took approximately 6 hours, while for the smallest datasets (a few hundred samples), it took under 10 minutes. Finally, we observed that when the original tabular data contains only a few dozen columns, adding 768-dimensional embeddings can increase the training time of gradient-boosted trees by up to a factor of 8.

*Implementation:* All our code is written in Python 3.8.2. We trained all models using one Intel Xeon Platinum 8260 or Intel Xeon Gold 6248 CPU and GPU. We conducted all of our predictive experiments using the XGBoost (Chen & Guestrin, 2016) library from Python. The Clinical-Longformer model is directly accessed from HuggingFace.

## 3.1 Patient Flow Predictions

We here evaluate the TabText framework on the same nine patient flow predictive tasks described in Section 2, but this time using a much larger data set that contains medical records of all inpatients over a four-year period. Each data point represents a patient day. There are 160 columns of different patient attributes on demographics, patient status, vital signs, laboratory results, diagnoses, treatments, and other information. We emphasize that this data set was not used for the development of the TabText methodology in Section 2. The summary of the data set and data sizes utilized can be found in the appendix in Tables 4 and 5. Experiments are reported across 10 random 75%-25% train-validation splits with a fixed test set. We use gradient-boosted tree models for all experiments performed in this subsection. The Tabular approach refers to gradient-boosted tree models trained on the tabular data, and the TabText approach refers to gradient-boosted tree models that include the TabText embeddings.

### 3.1.1 High Performance with Minimal Pre-Processing

We first leverage the TabText framework to replace the heavy lifting of data cleaning by simply creating a text representation for each data sample using the information as it appears in the raw data tables that we received from the hospital. In particular, columns that require data cleaning to convert to appropriate data types can instead be treated as free text—bypassing the need for extensive manual preprocessing typically required, as exemplified later in Section 3.1.2. For example, the sentence corresponding to a column for a sedation score with the value "-4 → deep sedation" can be considered as a categorical column and written as "sedation score is -4 → deep sedation", as opposed to parsing the original string into a numeric

value of -4 as part of the traditional pre-processing steps. Therefore, TabText representations enable us to quickly build baseline machine learning models utilizing the tabular data in its raw form. Only minimal

Table 1: Out-of-sample average AUCs achieved by baseline TabText models with minimally processed data and across 10 random train-validation splits.

| Prediction Task | TabText Baseline AUC |
|---|---|
| Discharge 24 hr | 0.803 |
| Discharge 48 hr | 0.790 |
| Enter ICU 24 hr | 0.801 |
| Leave ICU 24 hr | 0.839 |
| Enter ICU 48 hr | 0.757 |
| Leave ICU 48 hr | 0.835 |
| Expire 24 hr | 0.943 |
| Expire 48 hr | 0.933 |
| Mortality | 0.895 |

data pre-processing was required, including constructing the meta information of the tables and columns, which is estimated to have taken only a couple of hours of manual work. We then followed our TabText pipeline to train a gradient-boosted tree model for each classification task. The results are shown in Table 1. The baseline TabText models with minimally processed data already achieve high out-of-sample AUC performance, reaching practically implementable benchmarks in hospital systems. The average AUCs across 10 random 75%-25% train-validation splits are close or above 0.8 for all prediction tasks except for Enter ICU 48 hr, which is a notoriously difficult classification task (Na et al., 2023).

### 3.1.2 Enhanced Performance with Contextual Representation

Given the raw tabular data files, we perform several data processing steps following Na et al. (2023) to obtain a clean feature space. These steps include merging raw data tables, parsing string columns, encoding categorical variables, constructing features, and imputing missing data, as described in more detail below. In Na et al. (2023), the authors reported approximately one year of effort jointly by machine learning researchers and hospital specialists on obtaining, cleaning, and processing the data.

*String Parsing:* Some columns in string format require string parsing to extract numerical features as continuous variables. For instance, the normal ranges of laboratory tests in forms such as "50–70" are replaced with two columns: one with a value of 50 for the lower bound and another with a value of 70 for the upper bound.

*Categorical Encoding:* Categorical columns (e.g., department, mobility level, the reason for visit) must be converted to ordered numerical levels (consecutive integers) using label-encoding or binary categories using one-hot encoding. Due to the large number of categories, we use label encoding for all categorical variables.

*Feature Engineering:* To better capture the clinical information, we compute various auxiliary variables:

1) Current conditions extracted from records (e.g., whether the patient is in ICU or IV)

2) Normal indicators (whether the clinical measurement is within the normal/critical range) instead of the ranges themselves.

3) Counts (e.g., number of days in ICU, number of attending physicians).

4) Pending procedures/results (time until surgery, whether MRI is pending, etc.).

5) Historical record linked to the patient (e.g., number of days since the previous admission and length of the previous stay).

6) Non-patient-specific operational variables (e.g., day of the week, ward census and utilization, hospital admission volume on the previous day).

*Missing Data Imputation:* Since the raw data comes from a hospital system, it contains many missing values. We impute most missing entries with 0, except for a few cases. From communications with the hospital, we impute certain variables with prior knowledge of the meaning of missingness (e.g., missing Do Not Resuscitate (DNR) means the patient did not sign a DNR form). For some auxiliary variables, we apply some rules, such as imputing counts with 0 if no record exists and imputing the number of days since previous admission with a large number (e.g., 9999) if no previous admission exists.

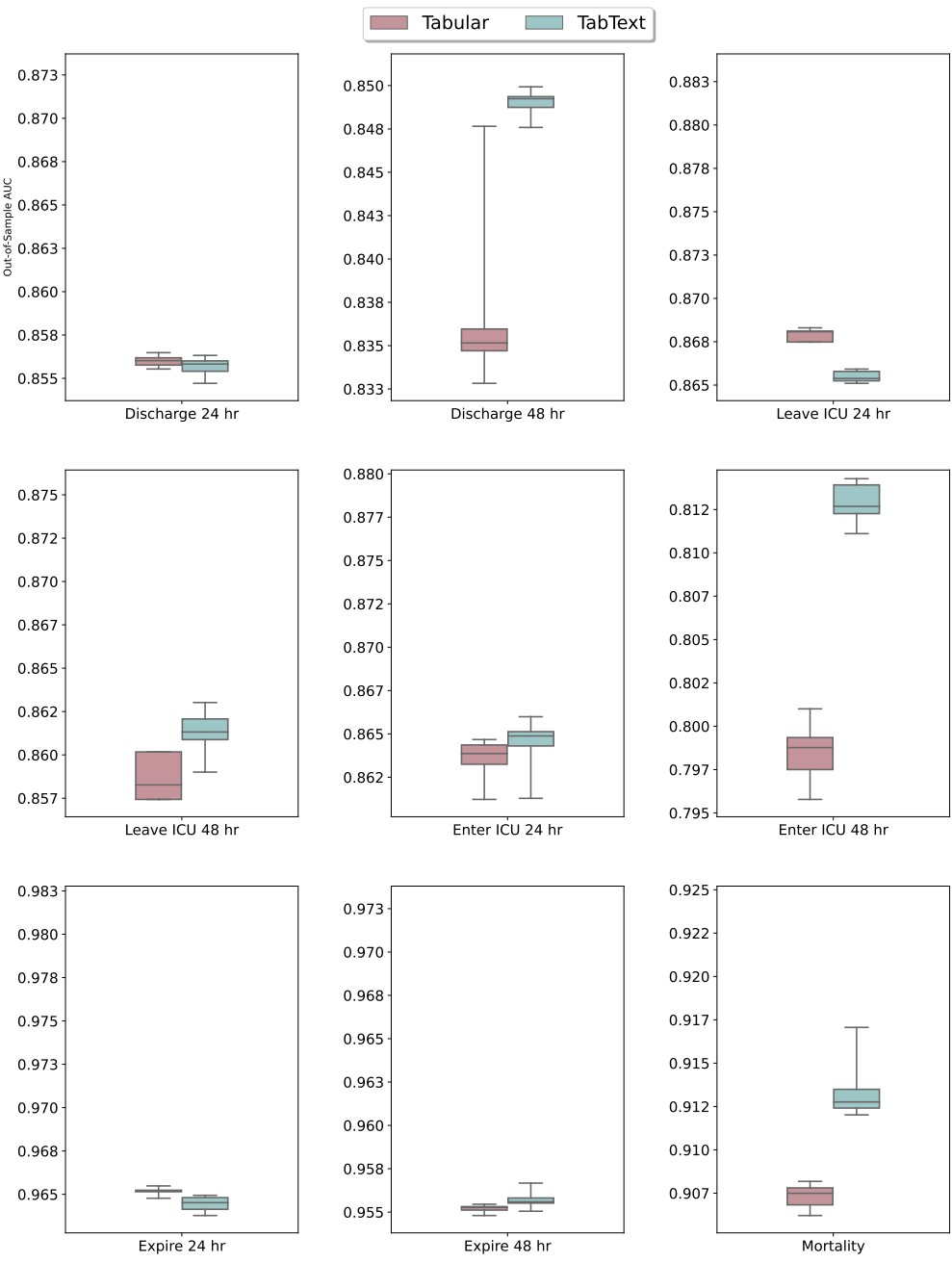

Figure 3: Boxplots for the out-of-sample AUCs across 10 random train-validation splits using Tabular vs. TabText models. We see that the largest TabText benefits occur for the classification tasks with high variability and low AUCs, while practically no effect was observed for the tasks with low variability and high AUCs.

We next process the data and group all features into 6 tables (see a summary in Table 4), which we combine and feed into the TabText Framework from Figure 1. We perform experiments on the same data and classification tasks as in Section 3.1.1 but using the cleaned data this time. The results obtained using the standard Tabular approach (XGBoost trained using clean tabular data only) and our TabText framework (XGBoost using the concatenation of both the clean tabular data as well as the Tabtext embeddings) are shown in Figure 3. The average AUCs across 10 random 75%-25% train-validation splits for the Enter ICU 48 hr and Discharge 48 hr prediction task are improved by an additive increment of 1.2%–1.4%. We also see a substantial but smaller benefit for Mortality risk prediction. For the remaining tasks, Tabular and TabText achieve similar performance with differences in average AUC smaller than 0.25%. We also notice in Figure 3 that the largest TabText benefits occur for the classification tasks with the lowest Tabular performance (high variability and low AUCs), while practically no effect was observed for the tasks with stable Tabular results (low variability and high AUCs).

### 3.1.3 Larger Benefits for Harder Predictions

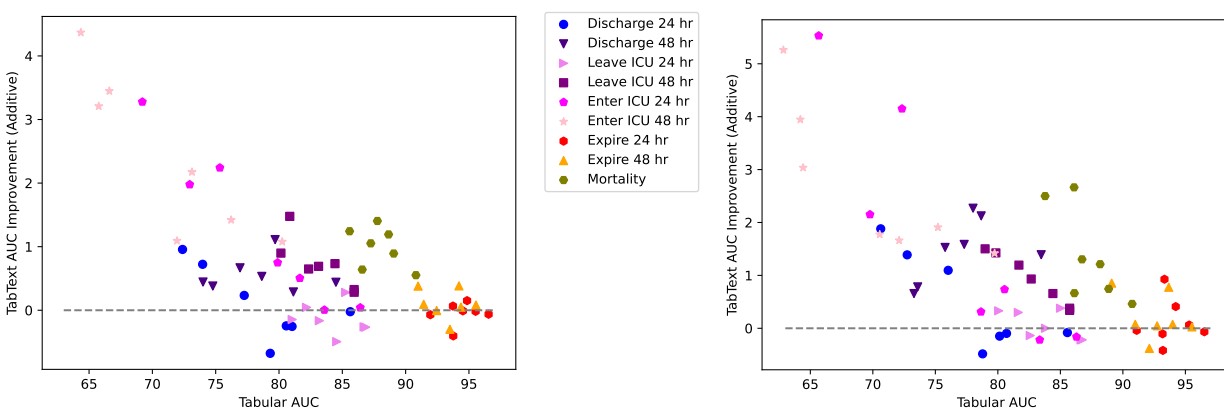

(a) Average out-of-sample AUC improvement at varying data sizes.

(b) Worst-case out-of-sample AUC improvement at varying data sizes.

Figure 4: TabText AUC improvement over the standard Tabular approach at varying data sizes. We observe that the improvement of TabText is most prominent when standard Tabular models do not perform well. For high-performing tasks, the advantage is less pronounced.

To better understand the regimes in which TabText provides the largest improvements in AUC performance, we repeat this experiment using smaller and larger training data sets. For each prediction task, we consider the original training data size as well as smaller data sizes (ranging from 2000, 3000, 5000, 10000, 25000, and 50000 patient days). We plot in Figure 4a (resp. Figure 4b) the average (resp. worst-case) TabText additive AUC improvement across 10 random 75%-25% train-validation splits, where the x-axis corresponds to the average (resp. worst-case) AUC of the standard Tabular approach and the y-axis quantifies the additive improvement on average (resp. worst-case) AUC achieved with TabText. Each scatter point represents the result of a prediction task (denoted by legends) on one of the 7 different data subsets. As in Section 3.1.2, we observe larger improvements on the more difficult prediction tasks with Tabular AUCs below 85%. On easier prediction tasks, where Tabular models already achieve AUCs over 90%, the benefit of TabText is near or below zero. When the Tabular AUCs are less than 78%, TabText brings a positive improvement on all results, including several instances of additive improvement over 2–3%. This suggests more potential benefits of augmenting tabular models with TabText representations for tasks with low Tabular performance, such as challenging medical prediction tasks with a lack of difficult-to-observe risk factors or a small sample size.

## 3.2 Public Datasets

Lastly, we evaluate TabText performance using 10 publicly available datasets from the UCI Machine Learning Repository (Kelly et al., 2023), which are described in Table 2. For tasks that are not healthcare-related we evaluate the TabText framework exactly as described in our training methodology, except we use Longformer (Beltagy et al., 2020) instead of Clinical-Longformer. Finetuning was performed with 7 epochs and batch size 4 for all datasets. We compare three types of classifiers—FT-Transformer (FT) (Gorishniy et al., 2021), Logistic Regression models (LR) (Hoerl & Kennard, 1970; Tibshirani, 1996; Zou & Hastie, 2005), and XGBoost (XGB) (Chen & Guestrin, 2016)—under three different input settings: tabular features only (Tab.), text features only (Text) via TabText embeddings, and the concatenation of both (Tab.+Text). We did not include results for FT-Transformer using TabText embeddings, as the computational cost of this method increases substantially with input dimensionality, and preliminary experiments indicated low performance. We also note that FT-Transformer applies standardization and quantile-based transformations during preprocessing, which provides a natural point of comparison with the discretization steps used to obtain TabText embeddings.

Table 2: Datasets from the UCI Machine Learning Repository. The columns n, p, and k correspond to data size, data dimension, and number of classes in the prediction task, respectively.

| Dataset | Acronym | n | p | k |
|---|---|---|---|---|
| Breast Cancer (Zwitter & Soklic, 1988) | BC | 286 | 9 | 2 |
| Heart Disease (Janosi & Detrano, 1989) | HD | 303 | 13 | 5 |
| Thoracic Surgery Data (Lubicz & Kolodziej, 2014) | TS | 470 | 16 | 2 |
| Autism Screening Adult (Thabtah, 2017) | AS | 704 | 20 | 2 |
| Cirrhosis Patient Survival Prediction (Dickson & Langworthy, 1989) | CP | 418 | 17 | 3 |
| Blood Transfusion Service Center (Yeh, 2008) | BT | 748 | 4 | 2 |
| Congressional Voting Records (con, 1987) | CV | 435 | 16 | 2 |
| Student Performance on an Entrance Examination (stu, 2018) | SP | 666 | 11 | 4 |
| Glass Identification (German, 1987) | GI | 214 | 9 | 6 |
| Flags (fla, 1990) | FL | 194 | 30 | 8 |

Table 3 presents the average out-of-sample AUC using 10 random 60%-20%-20% train-validation-test data splits. The top section of the table corresponds to models trained with the tabular approach (they use tabular features only). Among these, FT-Transformer performs followed by XGBoost. Logistic regression performs slightly worse overall, though still competitive on datasets such as SP and CV. The bottom section of the table shows the results from TabText models. XGBoost shows consistent benefit from the augmented feature space created by TabText embeddings, achieving higher AUCs across most datasets when compared to its tabular-only counterpart. In addition, the BT and CP datasets highlight the large (positive and negative) effect that TabText can have for logistic regression. Overall, models using TabText embeddings achieve the best or tied-best performance on six out of eleven datasets (BC, TS, AS, CP, BT, and FL), with particularly substantial gains on BC, TS, and FL. These results emphasize the potential of TabText embeddings in leveraging textual information to enhance predictive accuracy for multiple classification tasks.

## 4 Conclusions

This paper introduces TabText, a novel framework for processing tabular data by converting it into a text representation that captures important contextual information such as column descriptions. Our experiments show that augmenting standard tabular data with our TabText representations can improve the performance of standard machine learning models across 19 classification tasks, with larger improvements observed for the more challenging tasks. In addition, we demonstrate the efficiency of TabText in simplifying data pre-processing and cleaning, offering an alternative and flexible pipeline for generating high-performing baseline models.

Table 3: Out-of-sample average AUCs across 10 random data splits. The first three models use only tabular inputs; the last four use TabText embeddings. Numbers in bold correspond to highest performance for the corresponding dataset.

| Model (Input) | BC | HD | TS | AS | CP | BT | CV | SP | GI | FL |
|---|---|---|---|---|---|---|---|---|---|---|
| *Tabular Approach* | | | | | | | | | | |
| FT (Tab) | 0.710 | **0.800** | 0.523 | 0.737 | 0.781 | **0.752** | 0.981 | 0.756 | 0.880 | 0.839 |
| LR (Tab) | 0.712 | 0.664 | 0.560 | 0.579 | 0.738 | 0.575 | **0.992** | **0.764** | 0.800 | 0.594 |
| XGB (Tab) | 0.712 | 0.719 | 0.564 | 0.732 | 0.778 | 0.733 | 0.989 | 0.747 | **0.903** | 0.815 |
| *TabText Approach* | | | | | | | | | | |
| LR (Tab+Text) | 0.706 | 0.664 | 0.567 | 0.645 | 0.738 | 0.648 | **0.992** | **0.765** | 0.800 | 0.812 |
| XGB (Tab+Text) | **0.761** | 0.751 | 0.579 | 0.739 | **0.792** | 0.728 | **0.992** | 0.749 | **0.905** | **0.884** |
| LR (Text) | 0.700 | 0.770 | 0.574 | **0.747** | 0.656 | **0.752** | 0.771 | 0.639 | 0.616 | 0.720 |
| XGB (Text) | 0.745 | 0.739 | **0.592** | 0.738 | 0.668 | 0.743 | 0.943 | 0.714 | 0.735 | 0.801 |

Our experiments reveal the potential of TabText for improving the performance of standard machine learning models, and there are several research directions to improve our framework further. For instance, TabText relies on using NLP models that could generate high-quality embeddings for the input text, which motivates the development of more LLMs pre-trained with domain-specific data. An important direction for future work is to evaluate the generalization and transferability of TabText embeddings across domains and tasks. In particular, investigating zero-shot or few-shot performance on unseen datasets would help assess whether the learned representations can be reused without retraining—enhancing the practical utility of the framework in real-world applications. Moreover, augmenting tabular data with TabText embeddings adds a layer of complexity to the interpretability of the model output, and developing tools to maintain the interpretability of the tabular data would be an interesting direction for future research. TabText is a general framework that can be particularly useful for difficult classification tasks, and we hope that this work motivates further research for leveraging language in general machine learning applications.

## Data Availability

The patient-level data sets used for patient flow predictions are not publicly available due to the sensitive and confidential nature of hospital data. However, the data sets used in Section 3.2 are publicly accessible. To support reproducibility, we provide the complete TabText framework code, along with the data and metadata necessary to replicate our results on the public data sets (`https://anonymous.4open.science/r/TabText-18F0` ).

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

## A  Appendix

In this appendix, we provide additional information on the Results section.
The following are the hyperparameters that we grid-searched to obtain the optimal XGBoost models in Section 3.1:

**XGBoost.**   We searched over:

- Number of estimators: {100, 200, 300},

- Maximum depth: {3, 5, 7},

- Learning rate: $\{0.05, 0.1, 0.3\}$,

- $L_2$ regularization parameter: $\{1e^{-2}, 1e^{-3}, 1e^{-4}, 1e^{-5}, 0\}$.

Regarding Section 3.2, the hyperparameter grids for each model were as follows:

**XGBoost.**   We searched over:

- Number of estimators: $n_{\text{est}} \in \{10, 50, 100, 200\}$

- Maximum tree depth: $d_{\max} \in \{3, 5, 7\}$

- Learning rate: $\eta \in \{0.01, 0.05, 0.1, 0.5\}$

- $\ell_2$ regularization strength: $\lambda \in \{0, 10^{-4}, 0.001, 0.01, 0.1\}$

**FTTransformer.**   We searched over:

- Batch size: $\{8, 16, 32\}$

- Learning rate: $\{5 \times 10^{-4}, 5 \times 10^{-3}\}$

- Whether to include continuous features (`keep_cont` $\in \{$`True`, `False`$\}$)

- Number of training epochs: $\{4, 8, 16\}$

- Number of quantile bins used for feature discretization: $\{5, 10, 50\}$

All other parameters were kept as in the original paper.

**TabNet.**   We searched over:

- Number of decision steps: $n_{\text{steps}} \in \{3, 5, 7\}$

- Feature transformer dimension: $n_d \in \{8, 16, 32\}$

- Attention transformer dimension: $n_a \in \{8, 16, 32\}$

- Relaxation factor: $\gamma \in \{1.0, 1.2, 1.5, 2.0\}$

**Logistic Regression.**   We searched over:

- Penalty type: $\in \{\ell_1, \ell_2, \text{elastic net}\}$

- Inverse regularization strength: $C \in \{10^{-5}, 10^{-4}, 10^{-3}, 0.01, 0.1, 1, 10\}$

- Elastic net mixing parameter (if applicable): $\alpha \in \{10^{-5}, 10^{-4}, 10^{-3}, 0.01, 0.1, 1\}$

Table 4: Summary of tabular data, which contains different aspects of a patient's admission stay from patient's high-level demographics to precise lab measurements.

| Table # | Table Meta Information | Example Columns |
|---|---|---|
| 1 | Lab values | Platelet, Sodium |
| 2 | Chart measurements | Respiratory rate, oxygen concentration |
| 3 | Counting statistics | Number of medications, number of orders |
| 4 | Current condition | Oxygen device, is in ICU |
| 5 | Historical patient record | Previous admission, previous length of stay |
| 6 | Non-patient-specific data | Day of the week, ward census |

Table 5: Data sizes (number of patient days) for training and testing sets across the nine patient-flow prediction tasks.

| Prediction Task | Training | Testing |
|---|---|---|
| Discharge 24 hr | 572,964 | 265,917 |
| Discharge 48 hr | 572,964 | 265,917 |
| Enter ICU 24 hr | 385,132 | 180,075 |
| Leave ICU 24 hr | 73,013 | 34,669 |
| Enter ICU 48 hr | 292,659 | 138,947 |
| Leave ICU 48 hr | 68,472 | 33,011 |
| Expire 24 hr | 572,964 | 265,917 |
| Expire 48 hr | 572,964 | 265,917 |
| Mortality | 572,964 | 265,917 |

