# OpenReview forum: "TabText: A Flexible and Contextual Approach to Tabular Data Representation"
_TMLR — Rejected by TMLR_

### Review · Reviewer_yzQ3 · 2025-04-17

**Summary Of Contributions:**

The authors propose TabText, in its essence a preprocessing/data-augmentation framework for tabular data that leverages pretrained LLMs to create embeddings from tabular input data.
Subsequently, the authors show that training XGBoost on the created embeddings compared to the raw tabular data improves performance.

**Audience:**

Yes

**Claims And Evidence:**

No

**Requested Changes:**

- Since the idea is quite general and relatively simple, I think that the paper needs very thorough experimentation.
  - A much broader and more thorough HPO to truly detect whether TabText improves performance
  - Test whether simple column embeddings, combined with the actual tabular data have the same effect
  - A more thorough comparison to [8] might be sensible. The cingle sentence "but this is generated via LLM
prompting and is therefore exposed to hallucinations." is not quite an analysis, i.e. the authors fail to show why this is a problem.


# Questions
In Figure 1:
- As the final representation that is fed to the tabular model, according to the figure you are using:
   - Raw tabular data + Tabular Vitals data (in which format?) + Text embedding
   - As I understand your text, however, I believe you are using as input only the generated embeddings. Please clarify.
      - If this is correct, why not using the raw tabular data and augmenting it with additional information from the textual embeddings, but excluding the raw tabular data from embedding generation?

---

[8] Zifeng Wang, Chufan Gao, Cao Xiao, and Jimeng Sun. Meditab: scaling medical tabular data predictors via data consolidation, enrichment, and refinement. arXiv preprint arXiv:2305.12081, 2023.

**Strengths And Weaknesses:**

# Strength
- The method is very simple and can be applied to a lot of data


# Weaknesses
- Hpo, i.e. the grid-search is so small that the results are not really conclusive.
  - Additionally, no significance tests, accounting for multi testing (Bonferroni, Benjamini-Hochberg was performed)
  - In it's current state the authors claim, that TabText truly improves performance is - in my opinion - not fully supported
- It is not tested, whether a simple column-name embedding has the same effect
- The literature citations are quite often not really correct. It is news to me that [1] shows that Transformer models often struggle to represent language with numerical data, or [2] efficiently process tabular data in the form of language
- Additionally, there has been quite a lot of research that in the tabular domain that is missing in this paper:
  [3] - The complete PFN/ICL arches are missing
  [4, 5] - Nearest neighbour ideas are missing as well
  [6, 7] - Tabular DL advancements are also missing
   - while all of them are not completely related, since this is still tabular DL they should be mentioned.


## Literature
---
[1] Gorishniy, Yury, Ivan Rubachev, and Artem Babenko. "On embeddings for numerical features in tabular deep learning." Advances in Neural Information Processing Systems 35 (2022): 24991-25004.
[2] Somepalli, Gowthami, et al. "Saint: Improved neural networks for tabular data via row attention and contrastive pre-training." arXiv preprint arXiv:2106.01342 (2021).
[3] Hollmann, Noah, et al. "Accurate predictions on small data with a tabular foundation model." Nature 637.8045 (2025): 319-326.
[4] Gorishniy, Yury, et al. "Tabr: Tabular deep learning meets nearest neighbors in 2023." arXiv preprint arXiv:2307.14338 (2023).
[5] Ye, Han-Jia, Huai-Hong Yin, and De-Chuan Zhan. "Modern neighborhood components analysis: A deep tabular baseline two decades later." arXiv preprint arXiv:2407.03257 (2024).
[6] Thielmann, Anton Frederik, et al. "Mambular: A sequential model for tabular deep learning." arXiv preprint arXiv:2408.06291 (2024).
[7] Gorishniy, Yury, Akim Kotelnikov, and Artem Babenko. "Tabm: Advancing tabular deep learning with parameter-efficient ensembling." arXiv preprint arXiv:2410.24210 (2024).

---

### Review · Reviewer_cYWU · 2025-04-17

**Summary Of Contributions:**

This paper proposes a novel method, **TabText**, for representing tabular data as natural language descriptions, enabling large language models (LLMs) to generate semantically rich embeddings suitable for downstream clinical prediction tasks. The authors design a suite of strategies to caption tabular data and support the concatenation of multiple table-derived descriptions per sample. TabText is evaluated on diverse tasks, including patient flow prediction, breast cancer recurrence prediction, and glass identification, and consistently outperforms traditional tabular prediction models.

**Audience:**

Yes

**Claims And Evidence:**

Yes

**Requested Changes:**

1. **Expand the experimental evaluation**: To better demonstrate the generalizability and robustness of the proposed method, include additional benchmark tasks from diverse domains beyond the current limited set. This is important for establishing the approach’s broader utility.

2. **Clarify and strengthen baseline comparisons**: Provide a detailed description of the current baseline referred to as “Tabular,” and incorporate comparisons against state-of-the-art models such as TabPFN [1], TabLLM [2], and TransTab [3]. A more comprehensive baseline suite will ensure a fair and informative evaluation.

3. **Report and analyze generalization potential**: Investigate whether the pretrained LLM embeddings for tabular data can generalize across domains or tasks. For instance, evaluate zero-shot or few-shot performance on unseen tables or domains to demonstrate transferability, which would enhance the method’s practicality and relevance.

4. **Discuss computational cost and scalability** : Since the method involves fine-tuning LLMs for each task, discuss the computational burden and scalability considerations compared to traditional tabular learning methods.

**Strengths And Weaknesses:**

**Strengths:**

1. The paper provides a comprehensive exploration of various strategies for transforming tabular data into natural language, culminating in a well-motivated and grounded method for leveraging LLMs on tabular inputs.
2. It demonstrates a practical pathway for applying LLM-based architectures to tabular prediction tasks, enabling the reuse of pretrained models in clinical and general-purpose tabular domains.

**Weaknesses:**

1. The experiments are limited to a small set of tasks, which may not sufficiently establish the proposed approach's general superiority.
2. The baseline comparison lacks clarity and breadth. The term for the baseline “Tabular” is used without a clear definition or methodological description, and additional competitive baselines, such as TabPFN, TabLLM, and TransTab [1,2,3], etc., should be considered for a more robust evaluation.
3. The performance improvements over the baseline are often marginal. Given that the method requires fine-tuning an LLM for each task, the practical utility is questionable without evidence of cross-tabular or cross-domain generalization. Demonstrating that the pretrained LLM embeddings can transfer effectively across tabular domains would significantly strengthen the work.

[1] Hollmann, N., Müller, S., Purucker, L., Krishnakumar, A., Körfer, M., Hoo, S. B., ... & Hutter, F. (2025). Accurate predictions on small data with a tabular foundation model. Nature, 637(8045), 319-326.

[2] Hegselmann, S., Buendia, A., Lang, H., Agrawal, M., Jiang, X., & Sontag, D. (2023, April). Tabllm: Few-shot classification of tabular data with large language models. In International Conference on Artificial Intelligence and Statistics (pp. 5549-5581). PMLR.

[3] Wang, Z., & Sun, J. (2022). Transtab: Learning transferable tabular transformers across tables. Advances in Neural Information Processing Systems, 35, 2902-2915.

---

### Review · Reviewer_ttcY · 2025-05-11

**Summary Of Contributions:**

* The paper introduces TabText, a framework to extract more complete data representations from tables using language. The authors show that using this framework achieves high baseline accuracy in certain tasks
* The paper also presents a methodology for deciding upon the best language representations of the original data in their framework, which is expected to produce best accuracy downstream.
* It also introduces a systematic approach to converting tabular data into contextual language representations using pre-trained LLMs. They also try fine tuning these LLMs for better performance. The authors demonstrate that augmenting tabular data with these text-based embeddings can significantly improve the predictive performance of standard machine learning models across various healthcare and some non-healthcare tasks.

**Audience:**

Yes

**Claims And Evidence:**

Yes

**Requested Changes:**

Address weaknesses. Points 1,2,3 would be critical for an accept recommendation while point 4 would definitely be a nice addition.

**Strengths And Weaknesses:**

### Strengths

1) Practical Impact: Simplifies the data preprocessing pipeline significantly, and seems to achieve good performance
2) Detailed Methodology: Clear and comprehensive description of language construction (Figure 2) strategies and means of evaluation

### Weaknesses

1) While the authors evaluate the model across a variety of table tasks, it is challenging to contextualize its performance relative to other proposed methods. It might be more informative to assess the model on datasets that have been previously used in the literature, allowing for a clearer comparison to past work. The authors could also expand the non-healthcare evaluation to more diverse datasets to improve confidence in the model.

2) Section 3.1.1
* “columns that require data cleaning to be converted to appropriate data types can be simply transformed into text” - The paper does not answer the question how is the usual cleaning more expensive? Additionally, the proposed methodology in the paper seems to be converting the data to its quantiles which is already a widely known method in data processing.
* It would have been better to compare the task with the usual cleaning methods in this section rather than reporting raw accuracy which does not provide much information about the goodness of the proposed method

3) Section 3.1.2
* The existing baseline seems quite simplistic and lacks comparison with any other known methods. For example, the paper is using LLM to get textual embeddings for downstream prediction using other models (like trees). Why not just use the LLM for the whole task? There is several past work in this regard like TabNet, TabTransformer, TabTransformer, FT-Transformer, .. etc.

4) Ablation Studies: The authors did a nice job in presenting their decision methodology in Figure 2. But I feel an ablation study on each of these decisions would provide a very clear picture on what is exactly improving the performance in the downstream task.

---

### Comment · Action_Editor_rFc8 · 2025-05-16
**Reviewer-Author disucussion phase**

Dear Reviewers and Authors

I thank the reviewers for submitting their reviews. Now, we are in the review-author discussion phase. I want the reviewers and authors to clarify any questions.

Best
AE

---

### Decision · Action_Editor_rFc8 · 2025-06-16

**Recommendation:** Reject

**Additional Comments:**

N.A.

**Audience:**

Yes

**Audience Explanation:**

Although this paper has the aforementioned shortcomings, the use of LLMs for incorporating contextual information from tabular data is timely and vital. Therefore, the findings of this paper are likely of great interest to the TMLR audience.

**Claims And Evidence:**

No

**Claims Explanation:**

This paper proposes TabText, a preprocessing and feature extraction method for tabular data that leverages contextual information using Large Language Models (LLMs). The authors claim that TabText is beneficial as a general-purpose preprocessing method for tabular data.

The ideas presented in this paper are promising. However, although the authors have made improvements to the manuscript in response to the reviewers' comments, the experimental evidence to support its claims remains insufficient to meet TMLR's standards, as pointed out by multiple reviewers.

The main concerns are as follows:
- Inappropriate Experiment Settings: Multiple issues have been identified, including inadequate hyperparameter optimization and improper experiment configurations (such as the absence of early stopping, omission of continuous features, and restrictive fixed number of epochs in FT-Transformer).
- Insufficient Baseline Comparisons: Despite specific requests from reviewers, comparisons with state-of-the-art models, including TabM, RealMLP, and other ICL models (e.g., TabPFN, TabICL) are missing. Therefore, it is challenging to claim the effectiveness of the proposed method objectively.
- Insignificant Results: The presented results do not demonstrate a clear advantage for the proposed method, whose existence may undermine its effectiveness. Furthermore, statistical significance testing could strengthen the validity of the experiment results.
- Limited Experimental Scope: Even after the revision that adds experiments using the UCI datasets, some reviewers still consider the current experimental scope to be limited. A more extensive evaluation across a diverse set of tabular prediction tasks and base predictor variants may strengthen the authors' claims. Alternatively, the authors could narrow their focus to a specific domain, such as healthcare, and provide a more in-depth and comprehensive analysis.

Due to these points, the claims made in the paper cannot be considered to be supported by convincing evidence at this time.